# SpeechCraft: A Fine-grained Expressive Speech Dataset with Natural Language Description

## ABSTRACT

Speech-language multi-modal learning presents a significant challenge due to the fine nuanced information inherent in speech styles. Therefore, a large-scale dataset providing elaborate comprehension of speech style is urgently needed to facilitate insightful interplay between speech audio and natural language. However, constructing such datasets presents a major trade-off between large-scale data collection and high-quality annotation. To tackle this challenge, we propose an automatic speech annotation system for expressiveness interpretation that annotates in-the-wild speech clips with expressive and vivid human language descriptions. Initially, speech audios are processed by a series of expert classifiers and captioning models to capture diverse speech characteristics, followed by a fine-tuned LLaMA for customized annotation generation. Unlike previous tag/templet-based annotation frameworks with limited information and diversity, our system provides in-depth understandings of speech style through tailored natural language descriptions, thereby enabling accurate and voluminous data generation for large model training. With this system, we create SPEECHCRAFT, a fine-grained bilingual expressive speech dataset. It is distinguished by highly descriptive natural language style prompts, containing approximately 2,000 hours of audio data and encompassing over two million speech clips. Extensive experiments demonstrate that the proposed dataset significantly boosts speech-language task performance in both stylist speech synthesis and speech style understanding.

## CCS CONCEPTS

• **Information systems** → **Multimedia content creation**.

## KEYWORDS

Speech-Language Dataset, Controllable Speech Generation, Automated Speech Captioning, Multi-modal Processing

**ACM Reference Format:**
Anonymous Author(s). 2018. SpeechCraft: A Fine-grained Expressive Speech Dataset with Natural Language Description. In *Proceedings of Make sure to enter the correct conference title from your rights confirmation email (Conference acronym 'XX)*. ACM, New York, NY, USA, 10 pages. https://doi.org/XXXXXXX.XXXXXXX

## 1 INTRODUCTION

The success of *multi-modal learning* [39] has boosted a swift resurgence in the development of speech-language models over recent years, encompassing improvements in speech synthesis and automated audio captioning. Large-scale text-to-speech (TTS) models (e.g., VALL-E [36], Natural Speech 2 [29]) and audio-to-text (ATT) models (e.g., SALMONN [32], Qwen [2]) not only excel in traditional tasks through a *zero-shot* fashion but also exhibit emergent behaviors. The utilization of vast quantities of high-quality labeled data in training plays a crucial role in facilitating these advancements [43]. However, existing research mainly concentrated on fundamental audio characteristics, e.g., producing intelligible speech [19, 27, 28] and classifying broad sound event [5, 20]. In contrast, the nuanced audio interpretation that delves into the finer details of speech, particularly the speaking style, remains less explored.

The *style* of speech encompasses not only prosody but also the speaker's identity, emotional undertones, contextual cues, and scenes related to the topic within an audio clip [35]. Current TTS systems lack the necessary flexibility for precise and disentangled control over speech style. Current audio captioning systems struggle to capture the finer nuances besides rough detection of '*a man is talking while a dog is barking*'.

Nevertheless, few open-source datasets provide extensive details on vocal characteristics and nuanced descriptions [13]. The limited scale of existing fine-grained datasets significantly impedes speech-language style research. Consequently, there is an urgent demand for data that characterizes rich and detailed vocal information with natural language, both for expressive *speech language understanding* (SLU) and controllable speech synthesis.

It is widely acknowledged that human annotation datasets are typically costly, time-consuming, and limited in scope. To tackle the trade-off between large-scale data collection and high-quality annotation, we develop an *automatic speech annotation system for expressiveness understanding*, which conducts an exhaustive analysis of unlabeled audio across various dimensions and generates customized natural human language descriptions. The system incorporates expert classifiers and sophisticated captioning models to identify multiple speech attributes. Notably, for the first time, we not only take into consideration the basic speaking properties such as gender, emotion and pitch, but also pay attention to the detailed prosodic characteristics including word emphasis and topic information. Leveraging the extraordinary abilities of LLMs in language comprehension and generation, we employed a fine-tuned LLaMA 2 [33] to integrate attributes into comprehensive and stylistic descriptions. The produced descriptions are tailored for each audio piece, as opposed to previous works that employ predefined templates to fill blanks with attributes. The customization significantly enhances the diversity and nuances of the descriptions, aligning them with the unique characteristics of audio clips.

**Table 1: Examples of the style descriptions in different corpora. Each type of highlight/underline represents a unique speech attribute. In addition to the existing labels such as gender, pitch, volume, speed and emotional tone, descriptions in SPEECHCRAFT involve properties of age, topic, emphasis, and transcript for the first time.**

| FSNR0[‡] [16] | NLSpeech[§] [41] | PromptSpeech [11] | TextrolSpeech [13] | SPEECHCRAFT (ours) |
|---|---|---|---|---|
| Seem sad | The tone of the shock question revealed the sad feelings. | A distressful male sound appeared in low volume | A heartbroken woman 's voice, almost a murmur , is high-pitched . | Reflecting on a topic in the fields of Health and Fitness , a sad youth with low pitch and normal volume states, "Well, you know, life is holistic, Dave." She speaks at a fast pace , signifying her sadness . |
| In a hurry | His voice grew more agitated , and his tone revealed an urge and urgency . | Men , low tone , said loudly and quickly | Speaking at a fast pace , the pleasing male sustains a regular pitch and energy . | Engaging in a conversation about portraits , a natural youth female with high pitch and normal volume speaks swiftly , describing:"All his portraits seem to proclaim what a gentleman he is, and how he fascinates women!", intensifying the articulation of "fascinates" . |

[‡] The FSNR0 [16] style tags are the officially translated version from Korean. [§] The NLSpeech [41] style prompts are the officially translated version from Chinese.

The proposed annotation system can encompass the most fine-grained attributes and the most diverse natural language descriptions available. Considering English and Chinese are the two most widely used languages globally, we applied the annotation system to four popular bilingual speech datasets, including AISHELL-3 [30], Zhvoice[1], LibriTTS-R [18], GigaSpeech-m [6]. This effort resulted in the creation of the largest open-source expressive speech dataset, named SPEECHCRAFT, which comprises over 2,000 hours of audio data and more than two million speech clips, as shown in Tab. 1. Experiments in speech-related tasks show that SPEECHCRAFT dataset significantly contributes to the advancement of speech-language multi-modal learning in both TTS and speech-to-text domains. It enhances the performance of expressive speech synthesis, enables precise control over speech emphasis through natural language, and equips automated captioning systems with a broader context understanding capability, allowing them to describe detailed speaking styles beyond mere speech event detection.

In summary, our contributions are threefold:

- We proposed an automatic speech annotation system that employs all-encompassing speech interpretation methods and a fine-tuned language model to cultivate highly descriptive comprehension of speech expressiveness.
- We proposed an open-source, large-scale bilingual dataset available for advanced speech-language learning with fine-grained and expressive descriptions of speech, named SPEECHCRAFT.
- Leveraging the vast potential of SPEECHCRAFT, we accomplished controllable speech synthesis with precise emphasis control, and automated captioning with detailed descriptions of acoustic properties and speaker identity for the first time.

## 2 RELATED WORKS

*Content* and *style* are two aspects of vital importance in speech audio formation. Numerous speech corpus datasets have been released, such as LibriVox-based corpus [14]; however, datasets annotated with speech styles are limited. Besides, existing datasets typically describe the style of speech using tags or templates. Such annotations may be insufficient in diversity and richness for training large-scale speech-related models.

---

[1]https://github.com/fighting41love/zhvoice

This section will review existing stylistic speech datasets. Additionally, we will discuss related work on captioning tasks, which is highly relevant to speech language understanding and speech-style captioning as discussed in Sec. 5.3.

### 2.1 Tag-based Speech Datasets

The attempt at style control between text and audio starts from the usage of speaker ID tags in traditional speech language datasets, for instance, the series of LibriVox-based corpus [14], The speaker ID supported voice cloning tasks with the timbre that appeared in the datasets and intrigued one-shot TTS tasks which transfer voice and prosody with a short period of reference audio.

Subsequently, the release of a wide range of emotional speech datasets and multi-modal datasets, such as TESS [8], SAVEE [34], IEMOCAP [4] and MEAD [37], e.g., the field of *emotional speech synthesis* and *speech emotion recognition* (SER). However, emotion is usually defined as a categorization task, with six basic elements fixed: HAPPY, SURPRISE, FEARFUL, SAD, ANGRY, NEUTRAL, while other emotion tags varied by datasets, such as DISGUST, CONTEMPT, GRATEFUL, etc. The lack of an acknowledged definition in emotion classification also indicates that classifying emotion in single-word categories is not enough to represent emotion.

### 2.2 Natural Language Stylistic Datasets

Guided by the idea of extending the accuracy of emotional description to capture the rich emotion in speech better, researchers started to conduct speech datasets with natural language style prompts. As the pioneer in expressive speech synthesis with natural language prompts, InstructTTS [41] recruited human annotators to describe the speech utterance by overall emotion, emotion level, and a complete sentence, to describe the emotion conveyed in speech.

Emotional description alone has gradually become insufficient to meet the demands for style control. It soon came to researchers' minds that the variety of style factors and the diversity in style prompt descriptions are two of vital importance in elaborating speech style, as shown in Tab. 2. In contrast to human annotators which is always a costly and size-limited way, impossible to achieve large-scale data, the strong power and wide usage of LLMs drive the rapid development of this task. PromptTTS [21] employed five

**Table 2: Comparison between stylistic speech datasets. The proposed SPEECHCRAFT dataset possesses larger scale and finer-grained properties.**

| Dataset | Size of Dataset | | | | Property of Dataset | | | |
|---|---|---|---|---|---|---|---|---|
| | #Duration | #Clips | #Speakers | #Labels | Language | Audio Source | Description Form | Open Source |
| FSNR0 [16] | 26h | 19k | 1 | 1 | KO | Internal dataset | Style tag | ✓ |
| NLSpeech [41] | 44h | - | 7 | 2 | ZH | Internal dataset | Human annotation | ✗ |
| PromptSpeech [11] | - | 28k | - | 5 | EN | AUD, TTS synthesized | Human annotation + LM rewrite | ✓ ‖ |
| PromptTTS 2 [21] | - | 20k | - | 4 | EN | AUD | LLM template | ✗ |
| TextrolSpeech [13] | 330h | 236k | 1,324 | 5 | EN | AUD, Emotional dataset | LLM template | ✓ |
| Audiobox [35] | >500h | - | - | 8 | EN | Internal dataset | Human annotation + LLM rewrite | ✗ |
| SPEECHCRAFT (ours) | 2,391h | 2,250k | >3,200 | 8 | EN + ZH | AUD, YOU, POD, Smart Agent | LLM customization for each piece | ✓ |

‖ Approximately 85% of the corpus in PromptSpeech [11] was not released.

style factors, as well as utilized SimBERT [31] to generate more style prompts with similar semantics based on human annotation, which can be regarded as the first introduction of language models to solve the diversity problem. In PromptTTS 2 [21], researchers further promoted a prompt generation pipeline with an SLU part and an LLM part to compose high-quality text prompts. The essence of the pipeline lies in developing diversity in vocabulary format and the reuse of ChatGPT template sentences for utterances with the same labels. MM-TTS [10] and TextrolSpeech [13] also adopted ChatGPT in generating various prompt templates. It is worth noticing that TextrolSpeech is the largest open-source stylistic speech corpus so far, comprising 236,220 pieces of style prompts. It collected and curated a series of emotion datasets, as well as involved the traditional TTS dataset as neutral emotion utterance.

Despite the effort made by predecessors to produce as many prompt templates, the concrete categories of each style property limit the summation of total style permutation, since the templates do not provide additional speech style information. Taking TextrolSpeech as an example, all descriptions are derived from five style factors including gender, pitch, speaking speed, volume, and emotion. The emotion factor has eight options while other factors have two to three options each, a total of 432 combinations. Similarly, PromptTTS 2 has 54 combinations and MM-TTS provided 48 combinations. As a result, two different audio utterances with the same labels and templates would become totally the same in text representation after the prompt programming. The key barrier is that the descriptions are created based on LLM's understanding and rewritten capability given barely the text attribute labels instead of real speech audio samples, thus LLM could not provide unique annotations for each speech segment.

Another highly descriptive style dataset is Audiobox [35]. It leveraged all kinds of miscellaneous details with human annotation and a quality assurance rating system to gather better alignment towards human hearing perception. Yet, the data was unavailable and the method is by no means universally applicable. Consequently, there is a need to develop an automatic method that transcends the template-based description generation approach, achieving tailored style descriptions for individual audio pieces.

## 2.3 Auto-captioning from Audio to Speech

Audio or speech captioning refers to the task of generating descriptive text that describes the sounds, speech, and other contextual information of audio clips. It bridges the gap between auditory content and natural language, improving the understanding of multimedia content.

There has been a wide range of audio captioning datasets for audio-language multi-modal tasks from human-annotated captions [7, 15] to online harvested datasets [23, 38]. Large-scale audio captioning datasets enable the development of more sophisticated automated audio captioning (AAC) models that can understand complex auditory scenes. AudioClip [12] and CLAP [25] employed contrastive learning paradigm to learn acoustic concepts from natural language supervision, equip pre-training representative models with better audio interpretation and surpassed a bunch of downstream audio tasks including retrieval [17], event classification [5], and captioning [15].

As to speech captioning, the application of natural language supervision in AAC casts light on captioning with speech emotion description. Despite the lack of large-scale speech description data, SECap [40] proposed a speech emotion captioning framework with vivid natural language captions based on a small-scale human-annotated internal dataset. It took the same encoder-decoder architecture with CLAP and further employed the Q-former strategy to better disentangle the emotion-related speech information from the general semantic features. We made a good application of SECap in our automatic speech annotation system to capture unlabeled audio data with detailed and unique descriptions in emotional tone. It provides us with partial stylistic information which enables LLMs to directly interpret from authentic audio features. Regrettably, to the best of our knowledge, current speech interpretation models fall short of addressing any stylistic dimensions beyond emotion in fine-grained natural language captions. Furthermore, there is an absence of a speech captioning model capable of describing the full spectrum of speech style within a complete sentence.

## 3 DESCRIPTIVE SPEECH INTERPRETATION

### 3.1 Overview of the annotation system

We proposed an automatic speech annotation system for expressiveness interpretation. It contains a three-stage data processing pipeline: data preparation for multiple sourced raw speech segments, property extraction with an all-aspect SLU framework, and customized rewriting with a fine-tuned LLM. The system can equip in-the-wild speech with detailed and diverse descriptions.

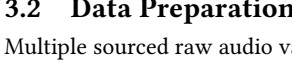

**Figure 1: System framework of the automatic speech annotation system**

## 3.2 Data Preparation

Multiple sourced raw audio varied in data quality and the form of original metadata. To reform the data with enhanced format, we first conduct data preprocessing to improve data quality, prepare data transcription, and establish standard metadata. All the speech segments, if not sourced from professional audiobooks, are fed into the speech enhancement system for audio quality improvement. Parallel to audio enhancement, the content of speech audio is transcribed using Whisper Large-v3 [26] if not provided. Other sundry items available in the original audio information such as title and raw description from the data uploader, and video category tags from the website, are mutually transferred by a language model to summarize the topic of the speech utterance.

## 3.3 Speech Style Recognition

The workflow of the speech style recognition is described as Fig 1. The speech is processed through various audio feature extraction models for characterizing speech in terms of its style properties. The output labels consist of pitch, energy, speed, age, gender, emotion description, and word emphasis, which are illustrated as follows.

**Signal Processing Tools.** We utilized traditional signal processing tools to analyze and classify audio signals and further predict acoustic properties such as pitch, energy, and speed. Speed and energy labels are categorized three-fold. As to the pitch label, following the precedent standard of Audiobox [35], we obey the common sense that female tends to have a higher pitch than males, setting categories of pitch by gender.

**Speaker Information Identification.** With the rapid development of large-scale audio representative learning methods, the high-level speaking features are well-detected by the hidden layers of powerful audio encoders. The pre-trained large-scale audio foundation models outperformed all kinds of downstream tasks compared to the respective task-specific models, including speaker features recognition and classification. Wav2vec 2.0 [1], attached

with some additional linear layers, are fine-tuned for identifying speaker information like gender and age.

**Emotion Caption.** Emotional tone is the fundamental key to style in speech. It is important to preserve the nuances of emotional tone in the original audio to the fullest extent possible. The audio utterances sourced from the audiobooks primarily consisting of storytelling narration by professional voice actors, exhibit minimal emotion. In line with TextrolSpeech, we deliberately set their emotional tones to 'neutral'. In the case of other emotion caption options, we take advantage of the most advanced emotion recognition technologies according to languages. As to English audio data, we adopt a self-supervised pre-training model for speech emotion representation called Emotion2vec. It provides convincing nine-class emotion recognition results and achieves SOTA on SER tasks. For Chinese audio data, we adopt the speech emotion captioning model SECap [40], which was trained on Chinese speech-caption data only. SECap helps describe fine-grained emotion in speech with diverse natural language. It captures the nuance of emotional cues in audio, including intensity and fluctuations, through short sentences rather than predefined single words. The customization of emotional tune makes the caption of each audio piece unique from others. The content serves as the textual basis for the final speech expressiveness description.

**Word Emphasis Detection.** Word emphasis plays a crucial role in speech expressiveness, conveying particular attitudes beyond the mere lexical content of what is spoken. Emphasis typically manifests as the strategic accentuation of certain words within a sentence. Therefore, the minimum unit for emphasis detection is set to be words in Chinese and English. Inspired by the lexical stress detection[2] in isolated English characters, we consider both the spectral and non-spectral features to hierarchically model the acoustic information. We model the spectral features with a residual convolutional neural network and non-spectral features with a deep

---

[2]https://github.com/LexicalStressDetection/lexical-stress-detection

neural network respectively. The details of model implementation can be found in the appendix. The emphasis of the sentence is the word predicted with Top 1 possibility.

## 3.4  Rewriting via LLMs

Capitalizing on the exceptional annotation capabilities of LLMs, we employed an expertise LlaMA 2[3] to take the group of attribute contents to generate natural language description for speech expressiveness. It is worth noting that we do not provide any structured formats for the description in advance to fill in the blanks as PromptTTS 2 does, but put emphasis on the richness of vocabulary and the accuracy in conveying the meaning of labels. To regularize LlaMA 2 to create promising results, we concentrated on detecting illegality and improving diversity during the fine-tuning progress. LLM may distort or omit content in the input that renders the output unusable, such as excluding certain labels or slightly altering the transcript. Irrelevant creative content including literary and contextual continuation between the multiple inputs also needs to be prevented. Therefore, we conducted thorough verification to abandon the low-quality generation. To better cultivate diversity and reduce the rough connection of labels instead of generating coherent sentences, three types of data enhancement methods were conducted, including order rearrangement of input attributes, expression synonym substitution, and multiple rounds of description translation. GPT-4 Turbo [24] is used to undertake a primary proportion of data and the fine-tuned LlaMA 2 undertakes the main role of rewriting.

Remarkably, we aimed to generate two versions of speech prompts to make the data more widely applicable. The speech description (denoted as the **Description version**) contains all available attributes regardless of the transcript. Besides, we involved speech transcript as an extra attribute to form a so-called speech instruction (denoted as the **Instruction version**). The primary motivation for involving transcripts in the text prompt can be described as the convenience for unified control tendency in the future development of speech interpretation.

- **Unified control of style and content.** Both style and transcript are significant in achieving unified control formation of speech audio in text speech interaction.
- **Unified control of global and detailed style information.** Previous models usually process transcript and style description through two channels respectively, resulting in difficulty in modeling the fine-grained instruction on designated words in transcripts. With the transcript contained in the description, it would be easier to model the global style, additionally, fine-grained instruction.
- **Unified control of speech and audio.** The unified construction form can be easily expanded from speech instruction to audio instruction datasets in the future, describing the overall scenario of sound events, speech content, and scene atmosphere.

## 4  SPEECHCRAFT DATASET

Deploying the annotation system on public speech datasets, here we introduce SPEECHCRAFT, a bilingual dataset for fine-grained and

---

[3]Due to the original LLaMA's limited proficiency in Chinese, we adopted an alternate version of LLaMA (Baichuan2-7B-Base [42]) trained on bilingual corpus.

expressive descriptions of speech. It is the largest open-sourced text description dataset in the dimension of both data scale and number of properties to characterize style. SpeechCraft encompasses a total of 2,108,710 pieces of speech descriptions for approximately 1,000 hours of audio data for each language. Due to the absence of public data with emphasis labels, we propose a method to generate speech with stress pronunciation on word emphasis with paired description to cater to word-level fine-grained style control. Furthermore, we conduct experiments to test the effectiveness of components within the data construction process in the section.

### 4.1  Data Sources

To facilitate further research in speech-language learning, we implemented the annotation system across large-scale speech datasets, including precise attribute labeling and expressive portrayals crafting. Considering a comprehensive set of factors such as dataset audio quality, the number and distribution of speakers, and the richness of emotional tones, we selected the Chinese AISHELL-3 and Zhvoice, along with the English GigasSpeech-m and LibriTTS-R, as our foundational datasets. Word clouds are illustrated in Fig. 3. See other detailed information and implementations of four datasets in the Appendix.

Notably, given the high-quality corpus, for instance, AISHELL-3 explicitly marked the auxiliary speaker attributes (gender, age group, native accents), some functionalities of the System were omitted in the actual construction of SPEECHCRAFT, adapting to local conditions.

### 4.2  Fine-grained Emphasis Speech Dataset

**TTS Backbone with Disentangled Feature Control.** Word emphasis is typically achieved through a combination of pitch variation, volume increase, elongation of sounds, and strategic pauses. All kinds of vocal cues work together to draw attention to the emphasized words. To better simulate the principles behind the generation of emphasis, we employed FastSpeech 2 [27] as the backbone due to its ability to generate high-quality speech with disentangled control over phoneme-level acoustic properties of pitch, energy, and duration. Adjusting the predictions of the acoustic features, we achieve loud volume, high pitch, and elongated sounds on the designed words. We tested on a mix of energy, pitch, and duration scaling factors to pinpoint the best combination that aligned with human audition.

**Emphasis Speech Generation.** As to the decision on emphasized words, we primarily assume the keywords of a sentence to be the reasonable candidate, and thereby conduct keyword extraction from speech datasets transcript. The FastSpeech 2 model underwent pre-training on AISHELL-3 and LibriTTS-R for the purpose of regenerating both datasets with keyword emphasis. Ultimately, we obtained 63,000 emphasized audio clips from AISHELL-3 and 75,000 from LibriTTS. Detailed implementation of Emphasis Speech Generation is illustrated in Appendix.

### 4.3  Validation of the Annotation System

The overall annotation system works by integrating detailed audio analysis with language model rewriting to create descriptions that

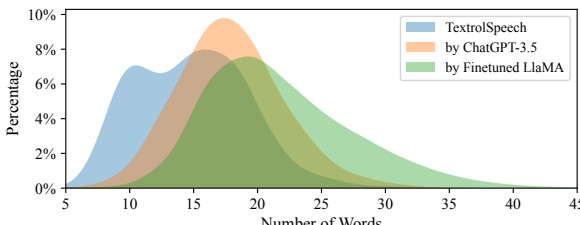

**Figure 2: Sentence length distributions of the speech descriptions generated by different models.**

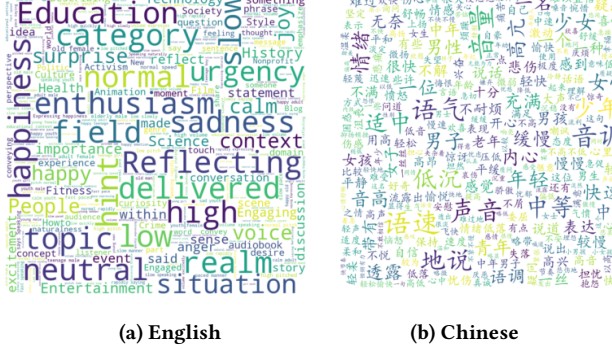

(a) English
(b) Chinese

**Figure 3: Word clouds of top 300 words in English and Chinese parts of SpeechCraft.**

can capture the nuances of speech expressiveness automatically. As a result, it is important to validate the accuracy of detailed attributes and the performance of the fine-tuned LLM in overall rewriting.

We first conduct an evaluation of the attribute predictors in style recognition on human-labeled data. The Wav2vec 2.0 [1] based age and gender classification model achieved precision of 97.72% and 87.7% respectively on the AISHELL 3 dataset which provided authentic speaker information. The officially released fine-tuned version of Emotion2vec [22] model for speech emotion recognition task outperformed other SERs with an accuracy of 84% on the internal English emotion dataset. To evaluate the performance of SECap [40] which describes speech emotion with short sentences, we involved ChatGPT to summarize the main emotional tendency of the captioned sentence within the range of single-word emotions. It turned out that the accuracy of the summary of SECap captions is 70.45% on a twelve-class internal Chinese emotion dataset.

Subsequently, we implement a series of assessments on the annotations of the fine-tuned LlaMA 2 with multiple dimensions to evaluate the effect of rewriting. We take the descriptions produced by GPT-3.5 Turbo [3] and TextrolSpeech as two sets of baseline. The **accuracy and completeness in preserving given labels** reflects the faith of the language model in carrying authentic information through the rewriting process. As shown in Tab. 3, the LlaMA 2 with instructional fine-tuning on data generated by GPT 4.0 had competitive results with GPT-3.5 Turbo [3]. Our annotation system's depiction outshone the earlier TextrolSpeech by reducing omissions or distortions throughout the transformation process, thereby guaranteeing the preservation of precise semantics embedded within a

broader range of intricate attributes. The **distribution of sentence length** in descriptions indirectly indicates that our longer sentences possess the potential to encapsulate a greater depth of detail, as shown in 2. Another interesting statistical discovery concerns the **position of spoken transcript among the speech instruction annotation**. The transcript exhibits probabilities of 5.55%, 33.05%, and 61.40% for occurring at the beginning, middle, and end of a sentence, respectively, showcasing syntactic diversity.

**Table 3: Assessment on rewriting of the fine-tuned LlaMA 2.**

| Prompt | TextrolSpeech | GPT-3.5 Turbo | Fintuned LlaMA 2 | |
|---|---|---|---|---|
| | EN | ZH | EN | ZH |
| Omission | 14.80% | 4.04% | 1.95% | 3.88% |
| Distortion | 2.10% | 6.10% | 7.50% | 6.14% |
| MOS | 3.58 | 3.84 | 4.02 | |

**Table 4: Data source distribution in SpeechCraft**

| Dataset | Language | #Duration | #Clips |
|---|---|---|---|
| SpeechCraft | EN + ZH | 2,381.54h | 2,249,579 |
| LibriTTS-R[†] | EN | 697.66h | 427,919 |
| GigaSpeech-m | EN | 739.91h | 670,070 |
| AISHELL-3[†] | ZH | 114.29h | 126,520 |
| Zhvoice | ZH | 799.68h | 1,025,070 |

[†] Statistics contain the original version and regeneratation with emphasis.

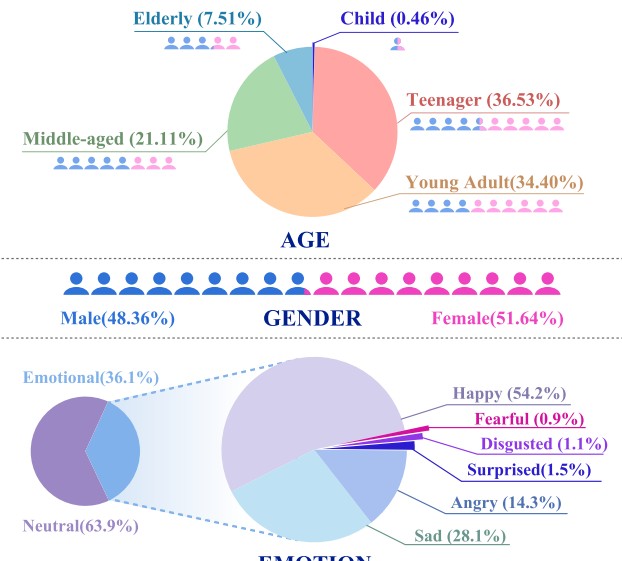

**Figure 4: Distributions of age, gender and emotion.**

Finally, we conducted a mutual experiment to evaluate both the quality of regenerated emphasis data and the effectiveness of the proposed emphasis detection model. We first trained models on the proposed regeneration of AISHELL-3 and LibriTTS-R respectively. The AISHELL-3-stressed achieves an accuracy of 88.55% on test set

**Table 5: Experimental results of expressive speech synthesis on the test set of TextrolSpeech.**

| Language | Dataset | TTS Quality | | Acc on Style Factors | | | | | | | MOS |
| | | MCD↓ | SECS↑ | Gender | Age | Pitch | Speed | Energy | Emotion | Mean_Acc | |
|---|---|---|---|---|---|---|---|---|---|---|---|
| EN | TextrolSpeech | 15.259 | 56.10 | 78.16 | 51.27 | 49.05 | 56.91 | 51.27 | 53.48 | 56.65 | 2.12 |
| | SPEECHCRAFT | **12.865** | **60.90** | **94.30** | **73.73** | **60.76** | **62.97** | **61.71** | **59.30** | **69.96** | 4.23 |
| ZH | SPEECHCRAFT | **11.200** | **61.50** | **94.74** | **57.19** | **41.05** | **58.60** | **58.25** | **79.23** | **66.09** | |

while the LibriTTS-R-stressed achieves an accuracy of 85.60%. The emphasis experiment on an internal dataset with human annotation on word emphasis is attached in the Appendix, demonstrating the effectiveness of our approach in modeling real-life stress patterns.

## 4.4 Data Analysis

The SPEECHCRAFT dataset is distributed evenly across both Chinese and English datasets, encompassing over 2,000,000 audio clips annotated with speech *Descriptions* and speech *Instructions*. The detailed information of the modified data in SPEECHCRAFT that sourced from public speech datasets are listed in Tab. 4.

Distribution of gender and age are shown in Fig. 4. The gender distribution is nearly balanced with a slight lean among different age groups. Most ages are distributed as TEENAGER, YOUNG ADULT, and MIDDLE-AGED, indicating a focus on individuals likely within the most active stages of life. English part of SPEECHCRAFT contains 36.1% emotional data mostly regarded as HAPPY, SAD, and ANGRY. The quantities of clips expressing SURPRISED, DISGUSTED, and FEARFUL are 5,626, 4,038, and 3,223, respectively, exceeding the scale of existing datasets. However, their proportions remain low in the context of the vast total amount of data. The unbalanced distribution of emotion resulting from in-the-wild data also indicates the frequency of real-life emotion tendency outside datasets.

## 5 BOOST PERFORMANCE VIA SPEECHCRAFT

To verify the impact of the proposed SPEECHCRAFT dataset, we conduct comprehensive experiments across various speech-language multi-modal learning tasks, including expressive speech synthesis, fine-grained emphasis control in TTS systems, and automated speech style captioning.

## 5.1 Expressive Speech Synthesis

The expressive speech synthesis task aims to generate high-quality speech audio under the intended speaking style in a seamless manner. Typically, the task employs a natural language description prompt as input to modulate the expressiveness of style. To validate the effectiveness of SPEECHCRAFT in replicating the intended expressiveness, we reproduce the text-controllable TTS model, Salle, and conduct training on both the original TextrolSpeech dataset and the proposed SPEECHCRAFT dataset for comparative analysis.

Salle embodies a VALL-E architecture with integrated natural language style prompts for refined speech generation. It adheres to the conditional codec language modeling architecture of VALL-E to produce the residual vector quantization [9] (RVQ) as audio representations. (See supplementary materials.)

As the TextrolSpeech dataset does not contain transcripts within its descriptions, we perform rigorous comparison experiments on

the *description* version of SPEECHCRAFT. We meticulously adhere to the official guidelines of Salle, training both models for 600,000 steps. Evaluation of the generated speech is conducted across three key dimensions: the recall accuracy of the style factors, audio quality, and user study. Attributes recall accuracy was assessed using a series of randomly selected labels with rewritten descriptions by GPT-3.5 Turbo [3], embodying a diverse array of all attribute dimensions. We obtain labels from the synthesized speech with the speech style predictors in Sec. 3.3 to evaluate the interpretation of the designated style description.

Traditional objective evaluation metrics, namely spectral entropy convergence score (SECS) and Mel-cepstral distortion (MCD), were utilized to assess the similarity of speaker and MFCC features between synthesized and original speech. This assessment was conducted using a small subset of the GigaSpeech-s TTS corpus. As to the user study, we conduct the mean opinion score (MOS) ratings from 1 to 5 on the overall naturalness of audio and its alignment with the style prompt.

As illustrated in Tab. 5, the extensive potential utility of large scale data is evidenced by the English subset of SPEECHCRAFT outperforming TextrolSpeech in all respects. However, the recall accuracy for pitch, energy, and speed in SPEECHCRAFT remained relatively low. This discrepancy was likely due to the regularization of inconsistent acoustic characteristics from different dataset domains under a uniform processing function.

## 5.2 Fine-grained Speech Emphasis Control

Utilizing synthesized emphasis data with paired fine-grained style instructions, we explore the potential applications of speech emphasis control by fine-tuning the expressive speech synthesis model. To assess the impact of incorporating transcript into the text description on fine-grained control capabilities like emphasis, we compare the emphasized speech effects between the *description* version and the *instruction* version of SPEECHCRAFT.

We evaluate emphasis accuracy in the synthesized speech in English and Chinese respectively. According to the design of the emphasis detection model in Sec.3.3, emphasis is detected by the unit of words. Therefore, word-level accuracy (Acc_w) denotes the nominal accuracy across all word pieces segmented from sentences. Sentence-level accuracy (Acc_s) refers to the true accuracy of correctly predicting an emphasized word based on the complete sentence Importantly, we include instructions without emphasis demands in the test set to evaluate the system's reliability through precise emphasis control. Sentences in this category are considered correct only if no word is incorrectly identified as emphasized. As illustrated in Tab.7, the proposed *instruction* version excels over the *description* version across all metrics, achieving 95.43% accuracy in

**Table 6: Experiments results of automated speech captioning on the test set of SECap** XY: Need better title.

| SECap | Captioning trained on SPEECHCRAFT |
|---|---|
| Felt happiness and joy. | A `young` `woman` , `voice high` , `pace swift` , revealed joy and delight in her emotion. |
| Appears to be very skillful. | A `young` `gentleman` , with an `elevated pitch` and `rapid speed` , articulated in `anger` . |
| The voice was full of curiosity, and the tone carried a careful anticipation. | A `young` `female` , with a `high-pitched` voice and a `moderate pace` , spoke with an air of confusion and misunderstanding. |
| Suspicious and puzzled about something. | A `young` `female` 's tone was `high-pitched` and the `pace was moderate` , speaking with a sense of doubt. |

**Table 7: The recall accuracy of emphasis detection. Abbr.: GT (Ground-truth), Des (Description), Ins (Instruction).**

| Lan | Version | Acc_w | Acc_s $R@1$ | Acc_s $R@2$ | MOS |
|---|---|---|---|---|---|
| | GT | 91.17 | 62.15 | 62.58 | 3.16 |
| **EN** | Des | 76.55 | 59.14 | 69.89 | 2.70 |
| | Ins | **88.53** | **87.77** | **90.96** | **3.98** |
| | GT | 86.97 | 63.60 | 64.25 | 3.79 |
| **ZH** | Des | 64.83 | 89.31 | 91.60 | 2.84 |
| | Ins | **84.02** | **94.06** | **95.43** | **4.05** |

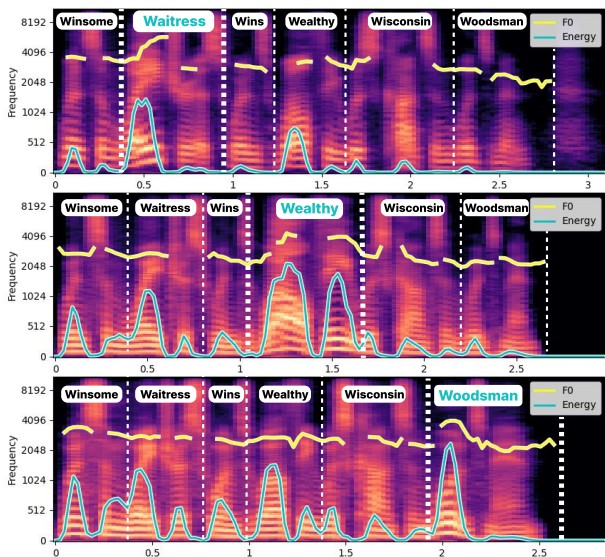

**Figure 5: Mel-spectrogram examples of speech emphasis control. Different stressed words on 'Waitress', 'Wealthy' and 'Woodsman' respectively with the same text instructions.**

Chinese sentences and 90.96% in English. This demonstrates conclusively that our data, enriched with emphasis word instructions and specific features of stressed pronunciation, fuses the fine-grained speech emphasis control with outstanding effectiveness.

**Case Study.** We further conduct a case study using a series of same base instructions varied only in the words emphasized, as illustrated in Fig. 5. The line of speech energy (green) shows a clear peak at the targeted words and aligns with the highest fundamental frequency (yellow) within the sentence. This distinct emphasis on the words highlights the flexibility and precision of our dataset's fine-grained, style-controllable synthesis capabilities.

## 5.3 Automated Speech Style Captioning

Automated speech style captioning goes beyond emotion captioning by providing comprehensive descriptions that encompass not only the emotional tone but also stylistic nuances such as acoustic properties and speaker identity. Owing to the nascent stage of speech style captioning development, we replicate the state-of-the-art automated emotion captioning model SECap (refer to Sec. 2.3), and retrain it on the *description* version of SPEECHCRAFT. Utilizing the semantic features extracted by the pre-trained audio encoder within SECap, the model achieves a surprisingly coherent interpretation of the overall speech style to a certain degree. For the first time, it was capable of capturing a descriptive sentence that included acoustic properties, speaker identity, and emotional tone. This achievement marks a significant advancement in the nuanced articulation of speech characteristics.

We conduct a user study to evaluate the detail and accuracy of the descriptions in capturing the audio style. Although our model marginally outperforms the original SECap (3.79 vs 3.58), there was consensus that speech style captioning, while capturing a broader range of audio characteristics, did so in a less expressive manner compared to SECap, which deals with nuanced emotional variations more explicitly.

**Case Study.** Some of the cases are documented in Tab. 5.2, highlighting the need for further research into the trade-offs involved in fine-grained speech style captioning regarding the scope and detail of dimensions.

## 6 CONCLUSION

In this work, we proposed an automatic speech annotation system for expressiveness interpretation that adopted various kinds of speech style recognition with LLMs rewriting to form detailed and customized descriptions for each utterance piece. Furthermore, we created SPEECHCRAFT, the largest open-source expressive bilingual speech dataset with natural language descriptions resulting from the annotation system, which has great potential in large speech-language model training. Experiments showed that our dataset strongly enhances the performance of both text-to-speech and speech-to-text models. It enhances fine-grained style controlment such as word emphasis in expressive speech synthesis and facilitates automated speech-style captioning in the broader field of speech comprehension and interaction technologies.

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

# A  APPENDIX

