# OpenReview forum: "SpeechCraft: A Fine-Grained Expressive Speech Dataset with Natural Language Description"
_acmmm.org/ACMMM/2024/Conference — MM2024 Poster_

### Official Review · Reviewer_6Tn3 · 2024-05-02

**Rating:** 3
**Confidence:** 4

**Summary:**

The author establishes a dataset called Speechcraft, which entails a significant trade-off between extensive data collection and meticulous annotation. Additionally, they propose an automated approach for constructing this dataset, primarily by labeling speech and subsequently utilizing a language model for concatenation. In comparison to previous similar dataset construction efforts such as Prompttts, Prompttts2, and Textrolspeech, the methodology does not introduce groundbreaking innovations. However, it enhances various aspects and generates a dataset with higher quality and finer granularity in style descriptions.

**Strengths:**

1. Compared to previous similar works, the automatic speech annotation system in Speechcraft demonstrates greater refinement. This is primarily achieved by incorporating textual content to ensure a one-to-one correspondence between style descriptions and speech segments. Furthermore, the introduction of age labels and prosodic emphasis enriches the details of style descriptions, expanding the dataset to approximately 2,000 hours. Additionally, the automatic speech annotation system is divided into the description version and instruction version, enabling the generation of even finer-grained style descriptions.
2. The article provides a clear exposition of the details involved in constructing the dataset, and the Speechcraft dataset itself is being prepared for open-source release. This initiative is poised to foster further advancements in text-to-speech (TTS) research, specifically in the realm of text style control, thus facilitating the overall development of the TTS community.

**Limitations:**

1. In terms of MM, innovation may be not enough, as many improvements in the automatic speech annotation system in this paper are more of a kind of prompt engineering. Previous works such as prompttts [1], prompttts2 [2], and textrolspeech [3] have further developed new corresponding controllable TTS systems. In addition, although the automatic speech annotation system has made further improvements in relevant details, its core still builds upon the aforementioned three works, rather than starting from scratch.
2. The main experimental results, as shown in Table 5, surprisingly exhibit an average accuracy of only around 60% for the controlled aspects. This indicates that achieving the desired controllable effects is challenging, as it faces a high error rate in both the textrolspeech and speechcraft datasets. Several factors could contribute to this, including the inability of the replicated Salle model to achieve the original performance and inaccurate labeling (whether marginal distributions should be discarded, for instance).
3. Considering the second point, in recent open-source contributions, ParlerTTS [4] has achieved promising results on the same task. It would be worthwhile to retrain the ParlerTTS model [5] using the speechcraft dataset to verify its effectiveness.
4. The lack of relevant ablation experiments raises concerns. Given that the main contribution of the automatic speech annotation system lies in prompt engineering, it would be advisable to conduct corresponding ablation experiments on each design aspect, as indicated in Table 5.
5. Taking into account the motivation behind building the automatic speech annotation system, speechcraft allows for longer style descriptions during testing. However, excessively long style descriptions might not be user-friendly. Furthermore, although age diversity has been considered, it may be worthwhile to further explore aspects such as accents and ethnicities. Additionally, while the author's consideration of style emphasis is commendable, achieving fine-grained control over different parts of a sentence, such as faster in the first half and slower in the second half, remains challenging. Speechcraft has made some progress in this regard, but significant advancements are yet to be made.
6. When constructing style prompts using language models such as GPT or LLAMA, a common issue is the diversity of style texts. For instance, when generating 1000 samples for a particular keyword label, there may be a phenomenon of homogeneity beyond the content text. I am curious about how the author has addressed this problem and I look forward to the author addressing my query and showcasing all style description texts from speechcraft during the rebuttal phase.
7. From a workload perspective, training a style-controllable TTS system with a dataset of 2000 hours (approximately 1000 hours of actual English data) may pose challenges. If speechcraft can further expand its scale to 10,000-20,000 hours while ensuring high-quality style descriptions and carefully balancing emotional relevance, among other factors, I would give further recognition to the work presented in speechcraft.



[1] Prompttts: Controllable text-to-speech with text descriptions


[2] Prompttts 2: Describing and generating voices with text prompt


[3] Textrolspeech: A text style control speech corpus with codec language text-to-speech models


[4] ParlerTTS: Natural language guidance of high-fidelity text-to-speech with synthetic annotations


[5] Open source models: huggingface/parler-tts


**I look forward to the author addressing my concerns during the rebuttal, and I will adjust my score (high or low) based on the feedback received in the rebuttal.**

**Suitability:**

3

---

### Official Review · Reviewer_37uT · 2024-05-10

**Rating:** 4
**Confidence:** 3

**Summary:**

There is a lack of large-scale stylistic speech datasets. In this paper, a large fine-grained speech dataset is proposed along with an automatic speech annotation system, which are proved reliable via validating the generated attributes and the rewritten corpus. Compared to previous datasets, SpeechCraft contains more types of annotations such as age, topic, emphasis, and transcript. The proposed dataset boosts the performance of several tasks such as expressive speech synthesis, fine-grained speech emphasis control, and speech captioning.

**Strengths:**

- The paper is in good writing and the literature review is comprehensive.
- The proposed dataset has the largest duration and more attributes compared to the previous dataset. It is the first bilingual stylistic speech dataset. A comprehensive analysis of the dataset statistics is presented, which shows that the data sources are very diverse. The annotation system is proven reliable by validating the extracted attributes and the rewritten corpus by LLM.
- The proposed dataset is beneficial for many tasks according to the experimental results in downstream tasks, and will make contributions to the community.

**Limitations:**

- In the experiment of expressive speech synthesis, how the accuracy of style factors are calculated? Besides, is there any other benchmark for style-guided speech generation? As the proposed dataset mainly focus on providing fine-grained descriptions, more experiments for proving the usefulness of the dataset in boosting the style-conditioned speech generation are better.
- Evaluation metrics for speech summarization are not clear. An explanation of how the accuracy is calculated is better.
- In addition, is there any subjective evaluation for captions generated by models trained on SECap and on SpeechCraft, respectively, as the accuracy is very close (3.79 vs 3.58).
- Involving transcription in the proposed dataset is one of the main difference compared to the previous dataset. Is there any experiments to show the benefits of involving transcription for downstream tasks?
- Whether the proposed dataset and the pretrained model for the three tasks will be open-sourced?

**Suitability:**

3

---

### Official Review · Reviewer_nFen · 2024-05-24

**Rating:** 4
**Confidence:** 3

**Summary:**

The authors propose a propose an automatic speech annotation system for expressiveness interpretation that annotates in-the-wild speech clips with expressive and vivid human language descriptions. With this system, authors create SpeechCraft, a fine-grained bilingual expressive speech dataset. SpeechCraft has about 2,000 hours of audio in more than 2 million bilingual clips.Using this dataset, the authors explored both tasks of controlled speech synthesis as well as automated speech style captioning, and Speech Craft brought a significant enhancement over previous work. SpeechCraft, as an open source, large-scale dataset, is promises to bring new hope to the field of speech synthesis.

**Strengths:**

- SpeechCraft processes audio through a series of expert classifiers and subtitle models, and with the help of LLM builds a high-quality automated speech annotation system capable of generating detailed and varied natural language descriptions for speech data and capturing nuances of speech expression.
- The authors have created SpeechCraft, a large-scale bilingual emotional speech dataset containing approximately 2 million speech clips and 2 million descriptive voice descriptions.
- Explored the role of natural language descriptions in facilitating tasks such as speech synthesis, fine-grained speech emphasis control, and automatic speech style captioning.

**Limitations:**

- Lack of a more detailed introduction to the use of large models to generate natural language descriptions, e.g. how to detect illegal behavior? What prompts are used to control the output of LLM? Providing more details contributes to a more comprehensive presentation of automated speech annotation systems, which is one of the core contributions presented in the paper.
- SpeechCraft has very well-developed property tags, but the dataset used in the article appears to be limited in terms of sentiment attributes, potentially introducing bias. Why not consider incorporating a proprietary emotion dataset to address this issue?
- Although SpeechCraft is a sufficiently large dataset, article novelty seems to be limited and the automated speech annotation system is more modeled after previous work. It would be helpful to enrich the articles if corresponding style-controllable baseline models could be further developed.

**Suitability:**

3

---

### Official Review · Reviewer_qjPt · 2024-05-26

**Rating:** 5
**Confidence:** 3

**Summary:**

This paper introduces a speech annotation system designed for expressiveness interpretation, which utilizes various speech style recognition techniques combined with large language models (LLMs) to generate detailed and customized descriptions for English and Chinese utterances. As the results, the work leads one expressive bilingual speech dataset with natural language descriptions, which could be helpful for the community and researchers.

**Strengths:**

The proposed automatic speech annotation system and the resulting SpeechCraft dataset, with its fine-grained and expressive speech data, enhances the performance of various speech-language tasks, making it a valuable resource for future research and development in the field.

**Limitations:**

(1) Many details are missing and not shown in the APPENDIX section, which could be important for readers to understand some data augmentation process. e.g how the Fastspeech2 system is adjusted for emphasis control.
(2) In section 4.2, for the emphasis of speech generation, have the authors evaluated the MOS or other quality of naturalness compared to real speech to understand the quality?

**Suitability:**

3

---

### Meta-Review · Area_Chair_n36N · 2024-07-10

**Recommendation:** Accept (Poster)
**Confidence:** 4

**Metareview:**

The authors present an automatic speech annotation system for expressiveness interpretation that annotates in-the-wild speech clips with expressive and vivid human language descriptions. First the speech audios are processed by a series of expert classifiers and captioning models and then by a fine-tuned LLaMA for customized annotation generation.  This system is used to create SpeechCraft, a bilingual expressive speech dataset. The dataset has been shown to boost speech-language task performance in stylist speech synthesis and speech style understanding.

Strengths:
- SpeechCraft is a large scaled dataset with fine-grained information that is lacking in prior work.
- Open sourcing of this data will help advance research in this field.

Weaknesses:
- Couple of reviewers have raised concerns around the lack of proper analysis and details about the LLMs being used in this approach. While some of this has been addressed in the rebuttal, it is a crucial component of the paper and warrants more coverage.
- There are also concerns about the extent of novelty in the approach beyond the automated speech annotation.

The authors should adequately address the limitations in the CR version.